# Molecular Characterization of pBOq-IncQ and pBOq-95LK Plasmids of *Escherichia coli* BOq 01, a New Isolated Strain from Poultry Farming, Involved in Antibiotic Resistance

**DOI:** 10.3390/microorganisms10081509

**Published:** 2022-07-26

**Authors:** Armando Hernández-Mendoza, Rosalba Salgado-Morales, Abimael Morán-Vázquez, David López-Torres, Blanca Inés García-Gómez, Edgar Dantán-González

**Affiliations:** 1Centro de Investigación en Dinámica Celular, Universidad Autónoma del Estado de Morelos, Avenida Universidad 1001, Cuernavaca 62210, Mexico; ahm@uaem.mx; 2Centro de Investigación en Biotecnología, Universidad Autónoma del Estado de Morelos, Avenida Universidad 1001, Cuernavaca 62210, Mexico; salgadomoralesr@hotmail.com (R.S.-M.); skin_abi@hotmail.com (A.M.-V.); lopez_5552@hotmail.com (D.L.-T.); 3Instituto de Biotecnología, Universidad Nacional Autónoma de México, Apdo. Postal 510-3, Cuernavaca 62210, Mexico; blanca.garcia@ibt.unam.mx

**Keywords:** phage-like, antibiotic resistance, lysogenic, insertion sequence, evolution

## Abstract

The increase in antimicrobial resistance has raised questions about how to use these drugs safely, especially in veterinary medicine, animal nutrition, and agriculture. *Escherichia coli* is an important human and animal pathogen that frequently contains plasmids carrying antibiotic resistance genes. Extra chromosomal elements are required for various functions or conditions in microorganisms. Several phage-like plasmids have been identified, which are important in antibiotic resistance. In this work, the molecular characterization of the pBOq-IncQ (4.5 kb) and pBOq-95LK (95 kb) plasmids found in the *E. coli* strain BOq 01, a multidrug resistant bacteria isolated from a poultry farm, are considered. Plasmid pBOq-IncQ belongs to the incQ incompatibility plasmid family and is involved in sulfonamide resistance. Plasmid pBOq-95LK is a lytic phage-like plasmid that is involved in the lysis of the *E. coli* BOq 01 strain and carries a bleomycin resistance gene and a strain cured of this plasmid shows bleomycin sensitivity. Induction of the lytic cycle indicates that this phage-like plasmid is an active phage. This type of plasmid has been reported to acquire genes such as *mcr-1*, which codes for colistin resistance and bacterial persistence and is a significant public health threat. A genome comparison, a pangenomic and phylogenomic analysis with other phage-like plasmids reported in the literature were performed to understand better the evolution of this kind of plasmid in bacteria and its potential importance in antibiotic resistance.

## 1. Introduction

Antibiotics have been used in animals for over 80 years in human and veterinary medicine for the treatment, prevention, and control of diseases, as well as growth promotion. They have also been shown to be essential for the sustainable production of beef and the control of animal infections that could be transferred to humans [1,2]. However, antibiotics have a wide range of collateral effects on microorganisms and the microbial communities in which they live, including antibiotic resistance, a phenomenon that is now recognized as one of the most serious global threats to human health at the present time [3,4,5].

It has long been shown that environmental, pathogenic, nonpathogenic, and commensal organisms contain functional antibiotic resistance genes [6]. *Escherichia coli* is an important commensal organism and a major pathogen; *E. coli* (ExPEC) is associated with extra-intestinal infections and causes the majority of urinary tract infections (UTIs) and sepsis-related bacteremia [7,8]. *E. coli* strains that cause diarrhea are grouped into six pathotypes according to the pathogenic mechanism: Enteropathogenic *E. coli* (EPEC), enterohaemorrhagic *E. coli* (EHEC), enterotoxigenic *E. coli* (ETEC), enter invasive *E. coli* (EIEC)*,* enteroaggregative *E. coli* (EAEC) and, diffusely adhering *E. coli* (DAEC) [9]. Pathovar Enteropathogenic *E. coli* EPEC attacks the epithelial cells forming adhesion and effacement (A/E) lesions, removing microvilli, and subverting host cell actin to form pedestals below the attachment site. These lesions cause a deficit in absorption and alter the electrolyte balance causing diarrhea mainly in infants, which can be lethal. EHEC attacks with the mechanism of pedestal formation. However, the mechanism differs from EPEC in that Tir is not tyrosine phosphorylated by the host cell, and pedestal formation is Nick-independent. Enterotoxigenic *E. coli* (ETEC) is anchored to small intestinal enterocytes by colonization factors and an adhesin found in the flagella. Secretion of enterotoxin and enterotoxin cause activation of cyclic AMP leading to diarrhea. Enteroinvasive *E. coli* (EIEC) invades intracellularly colonic epithelial cells in which it replicates, destroying the colonic epithelial barrier and dysentery. Because EIAC carries a pINV plasmid, invasion is similar to Shigella spp. Infection. Enteroaggregative *E. coli* (EAEC) adhere in a “stacked brick”-like pattern to the surface of small and large intestinal enterocytes through aggregative adhesion fimbriae (AAF) that causes the stimulation of IL-8 production and forms a biofilm [10]. The secretion of virulence factors such as toxin (Pet), which is encoded by the pAA virulence plasmid, is a serine protease autortransporter that targets α-fodrin, disrupts the actin cytoskeleton and indices exfoliation. Diffusely adhering *E. coli* (DAEC) adheres on enterocytes of the small bowel forming a diffuse adherence pattern mediated by afrimbrial (Afa) and fimbrial adhesins. These adhesins interact with decay accelerating factor (DAF), causing DAF aggregation in intestinal and urinary cells, cytoskeleton disorganization, and elongation and damage of brush border microvillii [9].

The ability of certain pathotypes of *E. coli* to colonize important farm animals and survive in meat products makes these organisms a particularly common cause of foodborne infections [11,12]. *E. coli* strains may vary in their genetic components and contain a variety of interchangeable elements, including plasmids, transposons, pathogenic islands, and other mobile elements, most notably cryptic, active, and lysogenic bacteriophages [11,13].

Because of the widespread use of antibiotics, there is increasing selective pressure to preserve related genes for antibiotic resistance. Plasmid-mediated gene exchange between bacteria is important for bacterial adaptation and flexibility, and it has contributed significantly to the rapid spread of antibiotic resistance genes in bacterial populations [14].

For example, it is known that members of the incompatibility group Q (IncQ) family are distinguished by a unique strand-displacement replication mechanism that can function in a wide range of bacterial hosts including Gram-negative bacteria and Gram-positive bacteria, they are small in size (5–14 kb) and present a high copy number (∼10–12 copies/cell). Furthermore, the plasmids IncQ are highly mobilizable and thus very promiscuous [15]. In the case of phage-like plasmids, their functional significance to the host is not known. However, they may play an important role in the spread of accessory adaptive traits [16]. Most of their genes encode for phage proteins, such as tail fiber, phage DNA invertase, phage holin, etc., although some carry genes of virulence or antibiotic resistance [17]. The recent discovery in China of a plasmid in *E. coli*, carrying the colistin resistance gene *mcr-1*, indicates the emergence of truly pan-drug-resistant bacteria. The plasmid pMCR-1-P3 carrying the *mcr-1* gene was likely formed via homologous recombination from a prophage that was originally located in the *E. coli* genome [18], but it has also been found in other members of the Enterobacteriaceae family living in human, animal, food, and environmental samples around the world [5,19,20,21]. In livestock environments, the number of plasmids carrying *mcr-1* may be higher than expected [18], and knowledge of phage-like plasmids needs to be increased to understand how genes of antibiotic resistance have been integrated into this type of element.

Previously, we reported the draft genome of *E. coli* BOq 01 a bacterium isolated from a poultry farm [22]. In this study, we characterize two plasmids found in this bacterium. One of them is pBOq-IncQ, which has a length of 4513 bp. It belongs to the IncQ family and encodes a total of five open reading frames (ORFs). The *strA*-*strB* genes were discovered to be linked to the sulfonamide (SU) resistance gene *sulII*, as previously described in other plasmids of this family [14].

The other is pBOq-95LK, with a length of 95,980 bp; it belongs to the incompatibility group pO111 and encodes a total of 119 open reading frames (ORFs), most of which are related to phage-related protein genes. This plasmid has resistance to bleomycin, an anti-tumor therapy drug, and a strain cured of this plasmid shows an increased susceptibility to this drug. It is also involved in cell lysis.

We perform a pangenomic and phylogenomic analysis, as well as several genome alignments, in order to understand the evolutionary history of pBOq-95LK with other phage-like plasmids and their role in antibiotic resistance, comparing this plasmid with other phage-like components, including some carrying the *mcr-1* resistance gene.

We cured the pBOq-95LK plasmid to confirm its involvement in antibiotic resistance and showed that pBOq-95LK has phage sequences that induce cell lysis.

## 2. Materials and Methods

### 2.1. Antibiotics Susceptibility Test

*E. coli* BOq 01 was assessed for their antibiotic susceptibility testing on different antibiograms using the Kirby–Bauer technique through the disc diffusion method [23] following the Clinical and Laboratory Standards Institute guidelines CLSI 2017 [24]. The selection of antibiotics was performed based on the 6th review published by WHO (2018), which lists the classes of medically important antimicrobial agents used in human medicine for risk management of antimicrobial resistance due to non-human use [25]. In this work, fourteen antibiotics corresponding to nine classes of antimicrobial agents were used to assess the drug resistance pattern of this strain, these include ten critically important—ampicillin (AMP) 10 µg/disk, carbenicillin (CB) 100 µg/disk, gentamicin (GEN) 10 µg/disk, streptomycin (STR) 10 µg/disk, kanamycin (KAN) 30 µg/disk, hygromycin B (HYG) 50 µg/disk, trimethoprim (TMP) 5 µg/disk, nalidixic acid (NAL) 30 µg/disk, fosfomycin calcium (FOS) 200 µg/disk, enrofloxacin (ENR) 5 µg/disk and four highly important—tetracycline (TET) 30 µg/disk, cefalexin (CLX) 30 µg/disk, sulfadoxine (SDX) 250 µg/disk and chloramphenicol (CHL) 30 µg/disk which were placed on LB plates and incubated at 37 °C for 24 h.

### 2.2. Plasmid Analysis

Genomic DNA was obtained using the AxyPrep bacterial genomic DNA miniprep kit (Axygen, Corning Life Sciences, CA, USA), and 5 μg of genomic DNA was used for whole-genome DNA sequencing of *E. coli* BOq 01 which were sequenced with the Illumina HiSeq platform (2X300-bp) system and annotated by Rapid Annotation using Subsystem Technology (RAST) version 2.0 [22,26]. We used the Eckhardt method to detect plasmid presence in *E. coli* BOq 01, as previously described [27]. To prove the circularity of pBOq-95LK, plasmid extraction was performed with the ZymoPURE^TM^ Plasmid Miniprep Kit. The plasmid DNA was used as a substrate to amplify and sequence the missing region with the primers were designed from both the 5′ and the 3′ ends of Scaffold_12_pBOq-95LK with outward directionality, (5′→3′) GTCTGTCGACCTGCTTGATC and Scaffold_12_pBOq-95LK (3′→5′) CATTCATCACCGCCTGCACG. A 408 bp amplicon of standard PCR was further sequenced through the Sanger method on an Applied Biosystems 3130xl Genetic Analyzer.

### 2.3. Plasmid Curing

To confirm that the plasmid pBOq-95LK (95 kb) confers resistance to bleomycin, the *E. coli* strain BOq 01 was cured of the plasmid. Briefly, BOq 01 was grown for 12 h at 37 °C (120 rpm) in sterile Erlenmeyer flasks of 100 mL containing 20 mL of LB broth. The resulting cell culture was used to calculate the amount needed to start a new collection with an initial OD600 = 0.001. The cell culture was transferred to sterile Erlenmeyer flasks of 100 mL containing 40 mL of LB broth, adjusted with sodium dodecyl sulfate (20% *w*/*v*). Inoculated Erlenmeyer flasks were incubated at 37 °C (120 rpm) for 12 h. After 12 h incubation, serial dilutions of the cell cultures were streaked on LB agar dishes and incubated at 37 °C for 16 h. Selected colonies were picked up and total DNA was extracted using a ZR Fungal/Bacterial Kit MiniPrepTM (Zymo Research, Irvine, CA, USA) according to the manufacturer’s instructions. To confirm a plasmid-free segregant that has lost the plasmid, the extracted total DNA was loaded in a 1% agarose gel. The resulting gel electrophoresis was visualized in a UV Trans-illuminator, observing the absence of the 95 kb band corresponding to the plasmid. Additionally, a PCR reaction was performed to amplify the region of the type II toxin–antitoxin system death-on-curing family toxins Phd-Doc present only in the pBOq-95LK plasmid with the primers Oligo (5′→3′) CCATATGAGGCATATATCACCGGAAGAAC and Oligo (3′→5′) GAATTCACTACTCCGCAGAACCATACAATCTACG, producing a fragment of 380 bp. The presence/absence of the amplicons was confirmed on a 1% agarose gel to select plasmid cured colonies. For bleomycin susceptibility analysis, the *E. coli* BOq 01 and the plasmid cured strains were streaked in LB plates with bleomycin (20 µg/mL), nalidixic acid (20 µg/mL) and incubated at 37 °C for 24 h.

### 2.4. Pangenomic and Comparative Plasmid Analysis

For comparative genomic analysis, we obtained sequences that share a significant similarity to pBOq-95K reported in the literature and stored in the GenBank database (Appendix A). Multiple sequence alignment and a synteny analysis were performed using Progressive Mauve integrated in Mauve version snapshot_2015-02-25 [28]. Visualization of prokaryote plasmids as a circular image to determine genotypic differences between them, displaying informative comparisons of pBOq-95LK to the others, was conducted with a BLAST Ring Image Generator (BRIG) [29]. Phylogenetic inferences were carried out analyzing 41 genes present in the 100% of the sequences analyzed. The ten sequences were annotated with the Prokka software version 1.2 [30] and the gff files obtained were used in the Roary software version 3.11.2 to create a pan genome of the plasmids [31]; the 41 genes sequences presented in all the plasmids were then concatenated. These sequences were then transformed to the phylip format (http://evolution.genetics.washington.edu/phylip.html, accessed on 25 November 2018–v1.4.4 figtree) with the perl script convert_aln_format_batch_bp.pl. The JModelTest v 2.1.10 software [32,33] identified the GTR + I + G as the nucleotide substitution model. The PhyML v. 3.1 was used for a Maximum Likelihood phylogenetic inference, and the Bayesian posterior probability was used to evaluate the reliability of the trees, which was visualized using FastTree 2 integrated in (http://tree.bio.ed.ac.uk/software/figtree/, accessed on 1 January 2021). Phylogenetic analyses of RepA and RepL were performed using the UPGMA algorithm in the MEGA software with a bootstrapping value of 1000.

## 3. Results

### 3.1. Plasmid Identification in E. coli Str. BOq 01

The draft genome sequence of *E. coli* strain BOq 01, a multidrug resistant bacterium isolated from a poultry farm, has previously been reported [25]. This bacterium has a 50.89 percent GC content, a 4.6 Mb genome size, and the sequence suggested the presence of two plasmids that confer antibiotic resistance genes. In order to analyze the genomic composition of the *E. coli* BOq 01 strain, the presence of plasmids was analyzed on agarose gels using the method of Eckhardt [27]. This analysis shows that two plasmids are present, one with an approximate size of 3500 bp and another with a high molecular weight, which cannot be inferred using this method (Figure 1). To identify those plasmids in the *E. coli* BOq 01 strain genome, we uploaded the contig sequences obtained with Illumina sequencing in the PlasmidFinder server. This program predicted the presence of the two plasmids, one located in Scaffold_10_pBOq-IncQ (4 kb) and another in Scaffold_12_pBOq-95LK (95 kb).

Once the scaffolds carrying the plasmid nucleotide sequence were identified, we analyzed whether the plasmids sequences were complete and whether they were linear or circular. A double alignment was performed with the same Scaffold_10_pBOq-IncQ, in which an overlap of 127 bp was identified at both ends of the contig sequence, indicating that it was a closed circled plasmid. In order to map the plasmid appropriately, this overlapping sequence was then removed from the Scaffold_10_pBOq-IncQ to preserve only one sequence. For plasmid pBOq-95LK, we designed a pair of primers from both the 5′ and the 3′ ends with outward directionality of the Scaffold_12_pBOq-95LK to amplify a sequence of 408 bp. A PCR product of the expected size was observed, validating that this plasmid was circular and has been completely sequenced. We propose to name Scaffold_10_pBOq-IncQ and Scaffold_12_pBOq-95LK as pBOq-IncQ and pBOq-95LK, respectively.

### 3.2. Molecular Characterization of pBOq-IncQ

On the one hand, plasmid pBOq-IncQ was identified on the PlasmidFinder server as an extrachromosomal element of the IncQ family with a 99.1 percent identity, carrying five ORFs (Figure 1), three of which are involved in antibiotic resistance (the streptomycin-resistance: genes *strA* and *strB* (an aminoglycoside 3’-phosphotransferase and an amidinotransferase, respectively), and *sul2* (a sulfonamide-resistant dihydropteroate synthase)), according to server ResFinder [34] (Figure 2A). Antibiotic resistance analysis shows that *E. coli* pBOq 01 is effectively resistant to aminoglycosides (streptomycin 10 µg/mL and kanamycin 30 µg/mL) and sulfadoxine sulfonamide (250 µg/mL) (Table 1).

Plasmids of this family are transferred between a wide range of bacteria isolated from different environments and were isolated for the first time in Gram-negative bacteria and classified as the IncQ incompatibility group. The IncQ plasmids are considered to be broad host range plasmids [35].

### 3.3. Molecular Characterization of pBOq-95LK

On the other hand, the PlasmidFinder server identified plasmid pBOq-95LK as a member of the pO111_2 incompatibility type, designated from the entero-hemorrhagic bacterium *E. coli* serotype:O111:H-1128 strain isolated in Japan in 2001 from patients with sporadic cases of diarrhea and bloody stools [36,37]. This plasmid harbors 119 ORFs, coding for 51 bacteria-specific proteins, 32 phage-related proteins, 23 hypothetical proteins, nine putative proteins, three tRNA-coded genes and other DNA enzyme repair genes, membrane transport proteins, and other cellular factors (Figure 2B). This type of plasmid is classified as a phage-like plasmid due to the presence of a large number of virus-related sequences [38]. Interestingly, the ORF 114, reported using the RAST annotation server as phage baseplate wedge tail fiber connector (T4-like gp9), is identified in the BLAST server and in the Protein Data Bank RCSB (UniProt), as a bleomycin hydrolase. Bleomycin is a first-line anti-tumor antibiotic used clinically for the effective treatment of certain cancers in combination with other antitumor agents. This compound of the glycopeptide family causes cell death as a result of the breaking of multiple strands of DNA through a mechanism that is not yet fully understood [39,40]. The resistance to bleomycin can be mediated by three different types of proteins or mechanisms: (i) bleomycin hydrolases; (ii) bleomycin N-acetylation enzymes that inactivate bleomycin-like molecules; and (iii) bleomycin-binding proteins (BMLA and BMLT) that act by trapping the bleomycin-like molecules. To analyze the effect of the loss of this plasmid, the *E. coli* strain BOq 01 was cured of pBOq-95LK using growth in SDS. We isolated the *E. coli* BOq 130, a cured strain of this plasmid, which was analyzed for their growth in the presence of bleomycin, showing an increased susceptibility to this drug compared to the strain carrying the pBOq-95LK plasmid (Appendix A) and the presence or absence of the phd/doc system was corroborated by PCR (Appendix A). Both strains of *E. coli*, the BOq 01 strain and *E. coli* BOq 130, with and without plasmid, respectively, were exposed to ultraviolet light to induce the lysis process. Only the strain with the plasmid pBOq-95LK showed clear translucent halos, a typical characteristic of phage-derived lysis activity (Appendix A).

### 3.4. Pangenomic and Comparative Analysis of Plasmid pBOq-95LK

A number of phage-like plasmids are derived from the Myoviridae family of phages P1 or P7, closely associated with temperate coliphages. These prophages infect a broad range of enteric Gram-negative bacteria and are unusual in being commonly discovered as free circular plasmids stably maintained without chromosome integration in bacterial cells. Phage P1 is responsible for infecting and lysogenizing enterobacteria such as *E. coli*, being retained as an independent, low-copy plasmid within the cell [16]. Recent advances in massive sequencing have found that variants of this type of plasmid have been reported to be involved in increasing resistance to last-resort antibiotics such as colistin.

Sequences similar to pBOq-95LK were searched in the non-redundant nucleotide database of the NCBI/BLAST server, detecting several phage-like plasmids with a high similarity to pBOq-95LK including plasmid 1 (LT905089.1) from *Salmonella enterica* subsp. *enterica* serovar Typhi strain ty3-243; plasmid p12579_1 (CP003110.1) from *E. coli* O55:H7 str. RM12579; plasmid p91 (CP023381.1) from *E. coli* strain 127; and plasmid pMCR-1-P3 (KX880944.1) from *E. coli* strain IMP163. Plasmid pMCR-1-P3, a plasmid of type IncY isolated from poultry farms in China was one of the most closely related plasmids with pBOq-95LK in sequence identity [18]. This element is a phage-like plasmid assigned to the IncY group and encodes a total of 108 open reading frames. The importance of this plasmid is increased by the acquisition of the gene *mcr-1*, encoding for the last-resort antibiotic resistance, colistin (polymyxin E) [18]. This gene has been considered to be responsible for the increase in multi-resistant bacteria that are considered to be the potential focus of human-risk epidemic outbreaks. Another plasmid, pMCR_SCKP-LL83, that carries this gene was isolated from a clinical strain of *Klebsiella pneumoniae,* named SCKP83 (MF510496). This plasmid had a single plasmid replicon pO111 [5], similar to pBOq-95LK. It is known that this element is not self-transmissible, and it has a region of 90.9 kb described as an intact phage very similar to phage P7 of *E. coli* [5]. The same team reported the *K. pneumoniae* strain KLB08 plasmid sequence (MK112268) carrying the *mcr-1* gene. At present, several plasmids carrying a similar *mcr-1* have been found in *E. coli* in various parts of China, suggesting that these phage-shaped plasmids have circulated widely in China [41]. The authors suggest that plasmids like phage P7 are not restricted to *E. coli* and may represent new vehicles to mediate the spread between *mcr-1* species [41].

After a thorough search of phage-like plasmids reported as similar to pBOq-95LK in the NCBI database, we selected four additional plasmids for a total of nine, including the plasmids from *E. coli* and *K. pneumoniae* containing the *mcr-1* gene, to compare their preservation, evolution, and synteny.

#### 3.4.1. Comparative Analysis of Phage-like Plasmids and pBOq-95LK

To identify the location of the locally collinear blocks (LCBs) and to analyze the synteny of the selected plasmids, we used the MAUVE algorithm to compare pBOq-95LK against plasmids *E. coli* pS51_2, *E. coli* p91, *Salmonella enterica* Tiphy Plasmid 1, *E. coli* pFDAARGOS_433 p3 (CP023896.1), *E. coli* pO111_2, *E. coli* p12579_1 (CP003110.1) [42], *E. coli* pMCR-1-P3, *Klebsiella pneumoniae* pMCR_SCKP-LL83, and *K. pneumoniae* pKLB08. MAUVE recognizes that most of the LCBs preserved are homologous regions that belong to phage sequences. With plasmid pBOq-95LK, all plasmids have highly conserved sequences, with pMCR-1-P3 (IncY) and *S. typhi* plasmid 1 (pO111) being the plasmids with the most similar segments and also with the highest preservation of synteny (Figure 3A). Among all the plasmids analyzed, the ORF 114, encoding for a putative bleomycin hydrolase, is completely preserved, suggesting that these plasmids could be involved in resistance to bleomycin. There are also several unrelated sequences between these plasmids, including the *mcr-1* gene (Figure 3B). Downstream *E. coli* pMCR-1-P3, *K. pneumonia* pKLB08, and the *K. pneumonia* pMCR_SCKP-LL83 *mcr-1* gene, there is a highly conserved transposase, ISApl1, indicating that this region could be involved in the translocation of the *mcr-1* gene. Interestingly, the *mcr-1* genes harbored on these plasmids are not inserted in the same region (Figure 4, blue lines). In pMCR-1-P3, the *mcr-1* gene is located between two areas that encode phage protein genes, but in pMCR SCKP-LL is located in a region involved in the replication process and in pKLB08 in a section encoding diverse proteins different from the abovementioned.

#### 3.4.2. Phylogenetic Analysis of Phage-like Plasmids and pBOq-95LK

Then, we analyzed the phylogenetic relationships among all plasmids, comparing the replication proteins RepA. The UPGMA tree clearly shows two clades separating the IncY of the pO111 incompatibility plasmids, indicating a clear correlation with the replication origin (Figure 5A). However, when using the RepL protein, this correlation is not observed (Figure 5B). Recently, Matamoros S. et al. [43] determined the population structure of *E. coli* and of mobile genetic elements (MGEs) carrying the *mcr-1* gene. No genotypic clustering by geographical origin or isolation source was observed, but 90% of MGEs carrying the *mcr-1* gene were present in plasmid with incompatibility IncI2, IncX4, and IncHI2 types [39]. We found the *mcr-1* gene in plasmid with IncY and pO111 incompatibility types, suggesting a wider specificity for the origin of replication.

It has been found that a single copy of IS *Apl1* is capable of mobilizing the gene *mcr-1*; however, other elements are needed, such as the structure IS *Apl1*, Δ-IS *Apl1*, *mcr-1*, and *pho* Δ, and these must be inserted into the *ant1* gene encoding a putative anti-repressor [44]. There are phage genes surrounding this element, which include *repL* (encodes a replication protein), *kilA* (encodes a protein of putative host destruction), and *simB* and *simC* (both encode proteins for host immunity) [38]. Plasmid pBOq-95LK only presents the genes *repL*, *kliA*, and *ant1*; however, the mechanism responsible for generating the structure IS *Apl1*, Δ-IS *Apl1*, *mcr-1*, and *pho* Δ that could have involved recombination is not present [38].

#### 3.4.3. Pangenomic Analysis of Phage-like Plasmids and pBOq-95LK

Since the phylogenetic analysis using Rep proteins was unclear, we decided to perform a pangenome analysis to obtain the core of 41 proteins present in 100% of the plasmids. The concatenated protein sequences were used for a phylogenetic analysis and the result of the Bayesian analysis of the trees obtained using ML identified several clades, where pBOq-95LK (pO111) was grouped with *E. coli* pMCR-1-P3 and *E. coli* FDAARGOS plasmid 3 (IncY and pO111, respectively). This clade is grouped into a bigger clade including *E. coli* plasmid p12579_1 and *S. typhi* plasmid 1 (both poO111), as well as *E. coli* strain 127, plasmid p91 (IncY) (Figure 6). The remaining *E. coli* plasmids form another clade with *K.*
*pneumoniae* plasmid pMCR_SCKP-LL83. In this clade, only *E. coli* str. 2014C-3084 plasmid unnamed1 [CP027320.1] shows IncY incompatibility, and the rest belong to the pO111 type, suggesting that the acquisition of the *mcr-1* gene in the phage-like plasmids is a stochastic process unrelated to the incompatibility type or the similarity of the sequences (Figure 6). However, these mobile components should be taken into consideration, as they may be associated with the high mobility of resistance genes.

## 4. Discussion

In this study, we reported the presence and molecular characterization of two plasmids in *E. coli* BOq 01, pBOq-IncQ, and pBOq-95LK. Our results indicate that both plasmids are involved in multiple forms of antibiotic resistance for this bacterium.

The plasmid pBOq-IncQ carries some genes that enable the bacteria to withstand the presence of sulfonamides and streptomycin, two common antibiotics used in poultry farms. pBOq-IncQ is a plasmid that belongs to the *E. coli* incompatibility group Q family. These plasmids are small (between 5 and 15 kb), can replicate in many bacterial hosts, and are increasingly recognized as carriers of clinically important antibiotic resistance genes in human and veterinary isolates.

The other plasmid is pBOq-95LK, a phage-like plasmid. This type of plasmid could be responsible for (i) the rearrangement by recombination with sequences of the chromosome or other extrachromosomal elements that could be responsible for the acquisition of multiple drug resistant genes, and (ii) the possibility that plasmids are active phages in the lysogenic phase that could be induced to initiate the lytic cycle in the cell and favor the recombination and incorporation of new genes in the bacterial populations.

For a comprehensive understanding of their fundamental genetic structure and diversification, pBOq-95LK was compared with nine sequenced phage-like plasmids chosen randomly to serve as representatives for a comparative linear structural analysis. According to the pangenomic analysis, 45 genes are part of the core genome; 15 are considered hypotheticals, while the rest are part of plasmids and phage sequences.

The toxin-antitoxin system (pdh/doc) and the Cre recombinase were among the typical plasmid core functions we discovered. These plasmids additionally encode either a repA or a repB replicase, but not both. These genes are found close to the parA and parB genes, which are essential for plasmid partition. The phylogenetic analysis of the replicases indicates that plasmids harboring repA are IncY, and repB are p0111, as was previously stated [45].

We also identified many conserved genes involved in the phage lytic cycle (holins, terminates, tails, baseplate proteins), or linked to phage functions (those in the phage head, tails, and tube proteins), or the morphogenetic genes *pmgCLP*. ORF114 encodes a baseplate wedge tail fiber connector for Phage (T4-like gp9). A BLASTp analysis, however, reveals that this ORF encodes a cysteine peptidase known as bleomycin hydrolase. Bleomycin, a common anticancer drug used in chemotherapy, can be broken down by this peptidase. Bleomycin hydrolase is highly conserved in organisms ranging from bacteria to humans [46].

The *mcr-1* gene for colistin resistance was present in three of the chosen plasmids. The comparative genome analysis does not point to the presence of a particular sequence that would increase the recombination events needed to add *mcr-1* to these types of plasmids. Even though the pBOq-95LK plasmid lacks the insertion sequences associated with the *mcr-1* gene, it is possible that this plasmid could pick up antibiotic resistance genes in the future, mobilizing this resistance in its offspring, and becoming one of the plasmids with antibiotic resistance discovered elsewhere in the world. This scenario is highly likely because plasmid-mediated resistance spreads outside farms and hospitals, including in wild animals [47].

## 5. Conclusions

Dealing with the problem of bacteria that are more frequently discovered in livestock farms and are antibiotic-resistant requires greater caution. Given that this particular strain *E. coli* BOq 01, is multidrug resistant, it is evident that there is a long-term risk involved and that extreme caution is required, particularly when choosing the proper antibiotics for use in both human and livestock health. Promising advancements in bioinformatics and genomics have enabled tracking of the emergence of multi-resistant microorganisms to antibiotics. For a complete understanding of the issue, supporting whole genome sequencing initiatives and keeping an eye on this problem are essential.

## Figures and Tables

**Figure 1 microorganisms-10-01509-f001:**
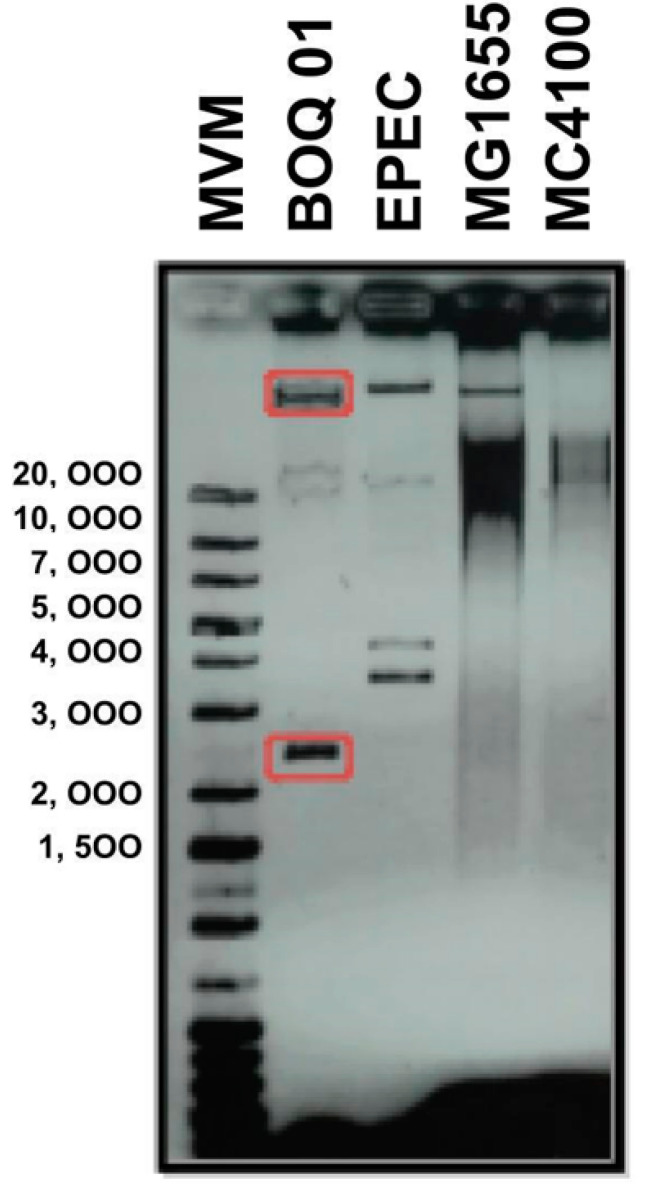
Eckhardt-type gel showing the plasmid patterns of *E. coli* BOq 01 strain and other *E. coli* strains. Lanes: (1) molecular weight markers (MWM), (2) *E. coli* BOq 01 strain, (3) enteropathogenic *E. coli* (EPEC), (4) *E. coli* MG1665 strain, and (5) *E. coli* K-12 MC4100.

**Figure 2 microorganisms-10-01509-f002:**
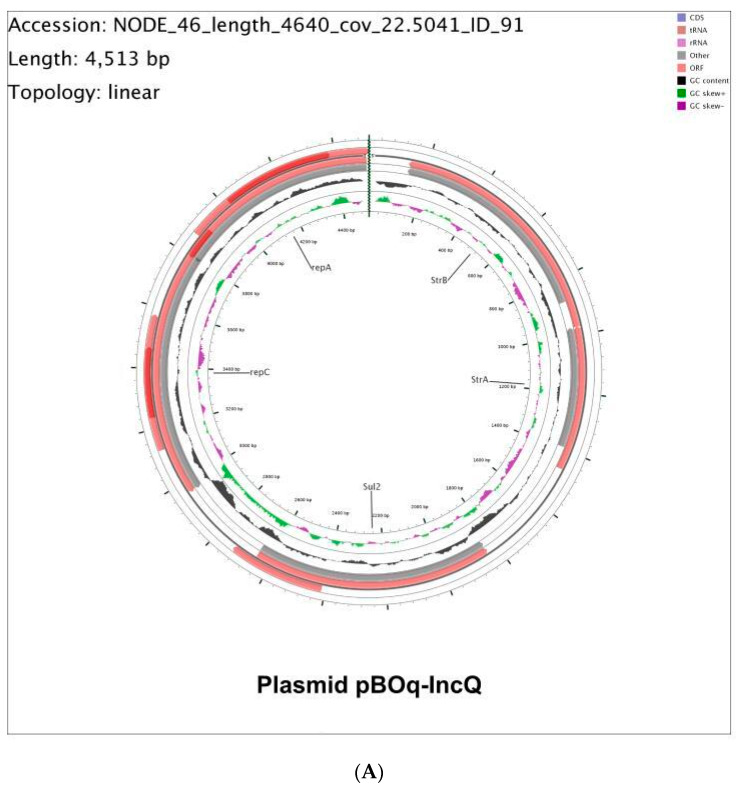
(**A**) Physical map of plasmid pBOq-IncQ from *E. coli* BOq 01 strain. (**B**) Physical map of plasmid pBOq-95LK from *E. coli* BOq 01 strain.

**Figure 3 microorganisms-10-01509-f003:**
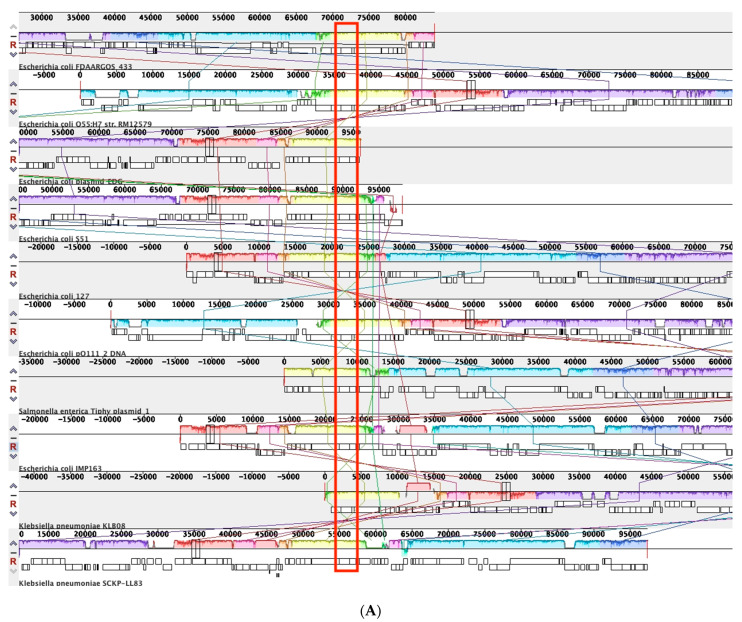
Physical map of a global alignment of phage-like plasmids against pBOq-95LK with MAUVE. (**A**) Alignment of plasmids centering at the position of bleomycin hydrolase (red rectangle). (**B**) Alignment of plasmids centering at the position of *mcr-1* gene (red rectangle) present in only three plasmids.

**Figure 4 microorganisms-10-01509-f004:**
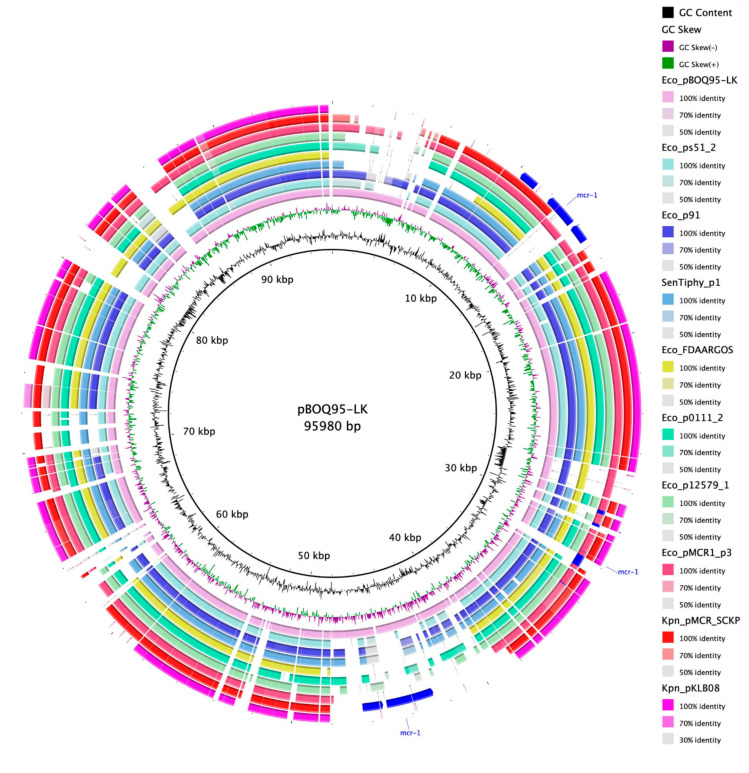
Circular comparison of plasmid pBOq-95LK (interior black line) with 9 phage-like plasmids. The BLAST Ring Image Generator (BRIG) program was used to align the plasmid pBOq-95LK with the other nine plasmids, and each sequence is represented in the concentric circles with color. The blue labels in the three external circles represent the location of the *mcr-1* gene encoding for colistin resistance and the mobilization elements around this gene.

**Figure 5 microorganisms-10-01509-f005:**
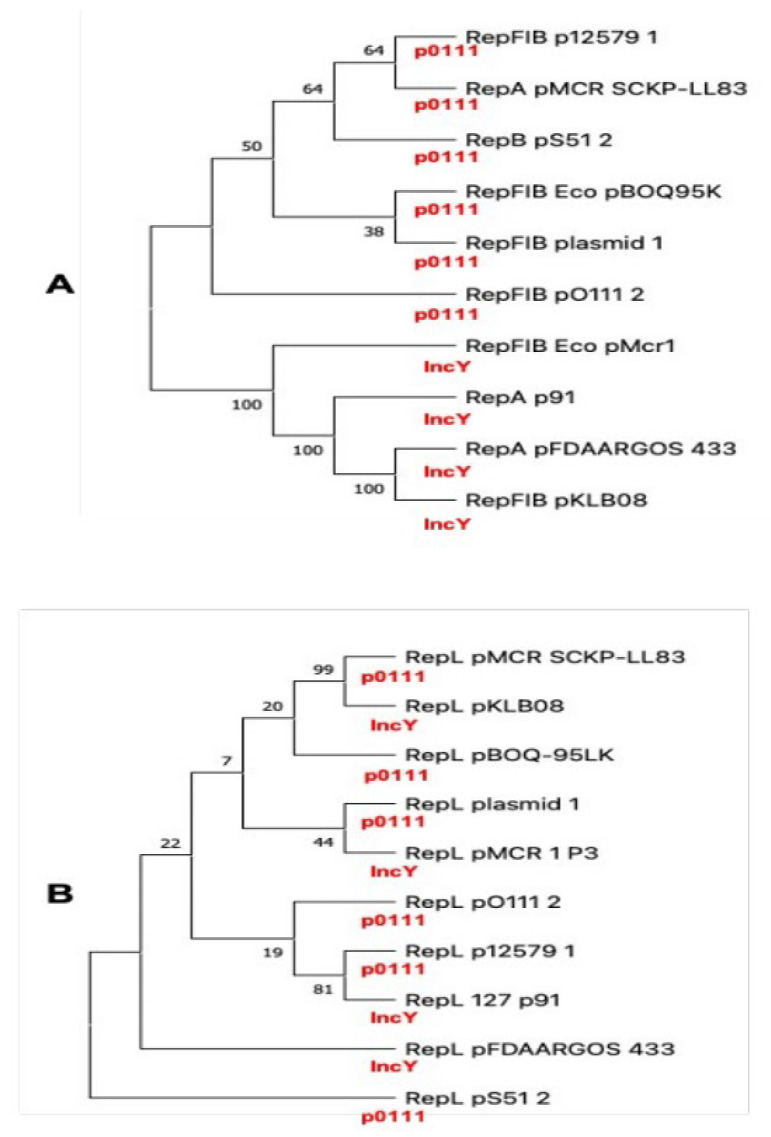
Phylogenetic tree using RepFIIB (replication origin, (**A**) and RepL (a protein involved in replication, (**B**), using UPGMA algorithm in MEGA software with a 1000 bootstrapping value).

**Figure 6 microorganisms-10-01509-f006:**
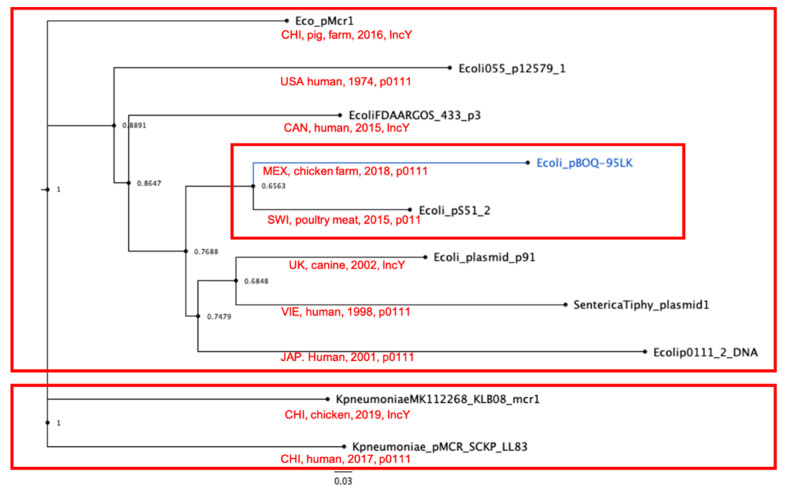
Phylogenomic trees of phage-like plasmids. Phylogenomic tree using 41 concatenated genes present in all plasmids (core plasmid genome) using Maximum Likelihood inference and the Bayesian posterior probability.

**Table 1 microorganisms-10-01509-t001:** Antibiogram test of the strain *E. coli* BOq 01 using 14 antibiotics grouped into 9 families.

Family	Antibiotic	Disk Content (µg/Disk)	Inhibition Zone Diameter (mm)			
			R	I	S	Results *E. coli* Strain BOq 01
Penicillins	AMP	10	≤13	14–16	≥17	6
	CB	100	-	-	-	6
Phenicols	CHL	30	≤12	13–17	≥18	8
Aminoglycosides	GEN	10	≤12	13–14	≥15	14
	KAN	30	≤13	14–17	≥18	13
	STR	10	≤15	12–14	≥11	6
	HYG	50	-	-	-	9
Tetracyclines	TET	30	≤11	12–14	≥15	10
Fosfomicyns	FOS	200	≤12	13–15	≥16	19
Quinolones and fluoroquinolones	NAL	30	≤13	14–18	≥19	16
	ENR	5	≤15	16–20	≥21	17
Cephalosporin	CLX	30	≤19	20–22	≥23	18
Diaminopyrimidine	TMP	5	≤10	11–15	≥16	25
Sulfonamides	SDX	250	≤12	13–16	≥17	6

Abbreviations: AMP = ampicillin; CB = carbenicillin; CHL = chloramphenicol; GEN = gentamicin; KAN = kanamycin; STR = streptomycin; HYG = hygromycin B; TET = tetracycline; FOS = fosfomycin calcium; NAL = nalidixic acid; CLX = cefalexin; TMP = trimethoprim; SDX = sulfadoxine; ENR = enrofloxacin. Interpretive results S = susceptible; I = intermediate or R = resistant.

## Data Availability

Not applicable.

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
