# Peer review of "Molecular Characterization of pBOq-IncQ and pBOq-95LK Plasmids of Escherichia coli BOq 01, a New Isolated Strain from Poultry Farming, Involved in Antibiotic Resistance"

_microorganisms, 2022, doi:10.3390/microorganisms10081509_

Round 1
Reviewer 1 Report
The manuscript by Hernández-Mendoza et al. is original, well written describing the molecular characterization of the pBOq- IncQ (4.5kb) and pBOq-95LK (95kb) plasmids found in the E. coli strain BOq 01, a multi-antibiotic- resistant bacteria isolated from a poultry farm. I support its possible publication after appropriate major modifications, as outlined below:
Line 17: please replace „antibiotic resistance” with „antimicrobial resistance” throughout the manuscript
Line 22: „we describe” – please avoid the using of personal verb forms
Line 23 „multi-antibiotic resistant” – please ensure that this is an appropriate term. If no, I suggest the using of „multidrug resistant”
Line 51: in order to increase the reader interest please complete the paragraph with the classification of the diarrheagenic E. coli strains according to the pathogenic mechanism into six sub-pathotypes
Line 79: I strongly suggest for the authors the using of a single term namely „susceptibility”, instead of „sensitivity” throughout the manuscript
Lines 87-88: the sentence is not fit for the introduction chapter
Line 94: “Fourteen antibiotics” – please explain the selection strategy of the used antimicrobials
Line 135: In order to validate their results, the authors need to refers to the used positive and negative controls within the conducted PCR reactions throughout the manuscript
Line 380: please delete „The antibiotics used are”
Line 453: the mean of “Supplemental Figure 1.” is not understandable, the figures and tables must be present I a supplementary material, and not in the main manuscript text body
Line 492: the discussion chapter must be substantially improved, ending with a conclusion chapter in which the authors must present the main conclusions, the study limitations and future perspectives in this research area
Reviewer 2 Report
Review of the article: Molecular characterization of pBOq-IncQ and pBOq-95LK plasmids of Escherichia coli BOq 01, a new isolated strain from Poultry Farming, involved in antibiotic resistance.
Manuscript ID: microorganisms-1815626
In my opinion, the proposed manuscript interesting and well prepared. I would recommend the article for publication in Microorganisms. Below I have presented several suggestions and remarks that authors should take into account preparing the final version of the article.
Detailed comments
Abstract – In my opinion this part of manuscript contains a lot of general, well known information (lines 17-22) whilst description of obtained results is very short (lines 22-28). Between lines 28-30 the authors presented short description of performed analysis but results of this part of the study are not presented in the abstract. Summarizing, the manuscript should be rewritten. The new version should contain aim of the study, short description of methodology and most important results.
Introduction
Lines 34-38 – antibiotics have been used in medicine for about 80 years not 60
Line 56 – explain the abbreviation IncQ
Line 58 – which plasmids?
Lines 58-59 – in my feeling this opinion is to general
Materials and methods
First of all at least short characteristic of E. coli BOq 01 should be presented (source and place of isolation).
Point 2.1. – some basic information should be provided – time and temperature of plates incubation
Point 2.2. – the method of DNA isolation should be mentioned.
Point 2.3. – what was the aim of isolation of colonies without the plasmid
Results – should be Results and Discussion.
Lines 196-198 – the authors used Kirby-Bauer method, thus the amounts of antibiotic in the disc cannot be presented as X µg/ml it is rather X µg/disc, and the criterion for classification of the strain tested as resistant (R), susceptible (S) or intermediate resistant (I) is diameter of growth inhibition zone.
Supplemental figures 1 and 2 should be includes in the final version of the publication.
Discussion – it this place the authors presented rather “Conclusions”. However, it is not well prepared, should be shortened.
Final decision – major revision
Round 2
Reviewer 1 Report
The authors correctly acknowledged all of the raised concerns!
Reviewer 2 Report
The authors have answered all my comments. In my opinion the manuscript can be accepted in current form.